# Concise, Practical Review on Transthoracic Lung Ultrasound in Prehospital Diagnosis of Dyspnea in Adults

**DOI:** 10.3390/medicina59020224

**Published:** 2023-01-25

**Authors:** Damian Kowalczyk, Wojciech J. Piotrowski, Oskar Rosiak, Adam J. Białas

**Affiliations:** 1Department of Pneumology, Medical University of Lodz, 90-419 Lodz, Poland; 2Department of Otolaryngology, Polish Mother’s Memorial Hospital Research Institute, 90-419 Lodz, Poland; 3Department of Pulmonary Rehabilitation, Center for Lung Diseases and Rehabilitation, Blessed Rafal Chylinski Memorial Hospital for Lung Diseases, 90-419 Lodz, Poland

**Keywords:** thoracic ultrasound, point-of-care ultrasound, POCUS, BLUE protocol, FATE, dyspnea

## Abstract

Ultrasonography is a relatively young but widely recognized method of imaging parenchymal organs, including the lungs. Our concise, practical review on transthoracic lung ultrasound (LUS) in the prehospital diagnosis of dyspnea in adults attempts to summarize current knowledge in the field. Furthermore, we discussed POCUS protocols in the analyzed context, discussing their usefulness. We concluded that bedside ultrasonography, or point of care (POCUS), is developing rapidly; however, the knowledge about the use of LUS in a pre-hospital setting is scarce, highlighting the need for further research in this field. Additionally, despite the possibility of using various ultrasound protocols in diagnosing a patient with dyspnea, there is no comprehensive and, at the same time, highly sensitive and specific protocol covering a satisfactory saccade of differential diagnosis of this symptom. It seems reasonable to conduct further targeted research to create such a dedicated solution.

## 1. Introduction

Ultrasonography is a relatively young, but widely recognized method of imaging parenchymal organs, including the lungs. Lung transthoracic ultrasound (LUS) is devoid of the risks associated with exposure to ionizing radiation posed by classical methods of chest radiology, i.e., radiography or computed tomography, and it is accessible and more widely applicable than such specialized methods of lung imaging diagnosis as magnetic resonance imaging. In addition, it is a method characterized by high sensitivity and specificity. For example, Lichtenstein reports a 97% sensitivity and 95% specificity for diagnosing oedema by assessing the presence of ultrasound artefacts. In the same study, he also notes that an A-profile evaluation with preserved pleural “sliding” sign accompanied by lower extremity thrombosis can correspond to the diagnosis of pulmonary embolism with a sensitivity of 81% and specificity of 99% [1].

In another study evaluating the effectiveness of diagnosing pneumonia during chest ultrasound imaging, Llamas-Álvarez A.M. et al. reported an 80–90% sensitivity and 70–90% specificity in detecting pneumonia. The study included 2359 patients [2].

A significant advantage of LUS is that the method can be used at the patient’s bedside or point of care (POCUS). POCUS is widely used in medicine, including emergency medicine, providing a valuable complement to the physical examination of the chest. Ultrasound also has a well-established position among diagnostic tools in emergency medicine in the setting of hospital emergency departments and anesthesiology and intensive care units (Figure 1). Of particular note, however, is the lack of studies on using LUS in emergency medical teams.

This manuscript aims to provide a narrative review of the latest protocols used in point-of-care lung ultrasound. A literature review was conducted with the Pubmed and Scopus databases using the keywords “POCUS”, “lung ultrasound”, “emergency ultrasound”, “paramedic ultrasound”, and “prehospital ultrasound”. The literature regarding point-of-care ultrasound of lungs was reviewed by the authors and presented in a narrative review style.

## 2. Differentiating Causes of Dyspnea Using Emergency Department Ultrasound

Respiratory and cardiovascular disease symptoms share many similarities; thus, their differential diagnosis is often a significant clinical challenge. One such difficulty is the diagnosis of dyspnea and the distinction between its cardiac and pulmonary causes. 

The emergency department patient population is a particular group due to the complexity of factors that can affect their clinical condition, as well as the need for immediate initial diagnosis, risk stratification, and decisions on further therapeutic management. The highly dynamic work environment, time pressure, and critical decision-making are long-known problems in the hospital emergency medicine [3]. 

LUS, which has an estimated sensitivity of 98% and specificity of 95.5%, appears useful in the differential diagnosis of the origin of dyspnea in patients diagnosed in the emergency department setting [4]. The study presented by Guttikonda et al. showed promising results, noting acute decompensated heart failure as the most common diagnosis with 97.3% sensitivity and 93.3% specificity in detecting LUS. A multivariate analysis found that jugular venous dilatation, fever, cough, ejection fraction, dilated inferior vena cava, and absence or reduced pulmonary sliding sign were independently associated with predicting cardiac and non-cardiac causes of dyspnea [5].

LUS seems to be a useful tool in the emergency department in the early detection of acute heart failure. Glöckner et al. described the eight-point method of lung ultrasonography in acute heart failure causing dyspnea in 102 patients undergoing lung ultrasound diagnostics in the HED. The sensitivity of LUS in this study in the diagnosis of AHF was 54.2%, and the specificity was 97.6% [6]. In another study, Sforza et al. described the role of bedside diagnostics with mobile ultrasound devices in emergency departments. The study concerned the differential diagnosis of dyspnea of cardiac and non-cardiac origin in the ER. The LUS showed a sensitivity of 92.6% and a specificity of 80.5% among 68 patients. It is worth emphasizing that the highest accuracy of the examination (90%) in the diagnosis of cardiac dyspnea was obtained thanks to the combination of LUS and POCUS ECHO [7].

After a review of the literature, the authors of this manuscript can conclude that knowledge about the use of LUS in a pre-hospital setting is scarce; however, it seems to be justified that this examination can also be performed in pre-hospital conditions with the use of mobile ultrasound devices. Making an early differential diagnosis in pre-hospital conditions may potentially facilitate the targeted diagnosis of a patient with dyspnea, and, based on this diagnosis, the implementation of appropriate treatment in pre-hospital conditions. As an example of such use of LUS, we can provide a study by Storti et al. who concluded that LUS has a central role, in the management of COVID-19 critically ill patients with acute respiratory distress syndrome, as a valid diagnostic and monitoring point-of-care technique, which could be performed in the emergency department, normal wards, intensive care units, but also in the prehospital setting [8].

## 3. Ultrasonography Point of Care (POCUS)

The dynamic development of medical technology has enabled the emergence of mobile instruments (Figure 2) that allow LUS to be performed at the bedside, in the patient’s home, in an ambulance, in the ED, or operating theatre. The development of the point-of-care concept has made it possible to perform immediate, bedside differential diagnoses of respiratory failure. Therefore, ultrasound, which has been an essentially hermetic area of interest for radiologists, is increasingly being used by specialists in other fields, i.e., doctors in various fields of internal medicine [9], anesthesiologists [10], or paramedics. 

An ultrasound examination is performed on patients who are in an immediate life-threatening situation. It is often associated with high stress and many factors that can negatively affect the examination. Fast pace, stress, dangerous situations, pressure from family members, and often from other medical team members do not facilitate the precise performance of the ultrasound examination. As a result, it is not uncommon for the person performing the LUS examination to have difficulty performing it and interpreting the images obtained. Simple ultrasound protocols are being developed to minimize such risk for use by other medical personnel who are not involved in radiology daily. POCUS evaluation is based on simplified examination protocols applied to the specific clinical context of the patient. The protocols were developed for the rapid ultrasound evaluation of life-threatening conditions, the goal of which is to identify disease or pathology that may threaten a patient’s health or life in the coming minutes or hours.

## 4. POCUS Teaching

Değirmenci et al. investigated the usefulness of a short course conducted on ultrasound simulators in teaching the Focused Assessment with Sonography for Trauma (FAST) protocol. The authors enrolled 60 participants in the study, including emergency medicine residents, medical interns, and paramedics—20 in each subgroup. Resident physicians achieved 99.5% image correctness and 94% correct diagnoses. Interns achieved 98.5% and 88%, respectively, and paramedics achieved 98% and 81.5%. Thus, in the study presented here, it seems that the level of theoretical knowledge did not affect the ability to obtain correct imaging. The differences were in the correctness of the diagnoses made in proportion to the extent of theoretical knowledge [11]. This study is the only report in the literature that considered a group of paramedics. 

Emergency medicine training in the pre-hospital setting in the US can be divided into two medical professions, i.e., basic—EMT (emergency medical technician) and advanced—EMS (emergency medical system). Both occupations are a form of specialized training based on post-graduate courses. The training and competencies of both professions vary by US state. However, the consistent assumption is that EMT is a form of a paramedic, and EMS is a form of EMT. Every US state has EMS schools on an advanced course basis (e.g., Boston EMS Academy), which lasts from 6 to 18 months depending on the school. Before taking the EMS course, an EMT course is required along with work experience (usually about 1200 h). After passing the EMS course, candidates take the NREMT state exam conferring the title of “paramedic”—the highest level of EMT certification in the US (comparable to paramedic in Poland). Tasks of a “paramedic” in the US include carrying out ALS activities, administering intravenous drugs and IV fluids, or advanced airway protection, including endotracheal intubation. The estimated period of education of a “paramedic” along with the necessary work experience (EMT and EMS) is 18–32 months. According to the decree of the Minister of Science and Higher Education of 26 July 2019, the process of training a paramedic in Poland consists of completing a three-year bachelor’s degree (six semesters), where the number of classes is a minimum of 3675 h, including 960 h of professional practice [12,13,14,15]. 

As part of a course designed to prepare students for the FAST examination, Damewood and colleagues [13] compared the effectiveness of a multimedia ultrasound simulator with classical human models. A total of 92 students were divided into two groups: the multimedia simulator team (*n* = 44) and the human model team (*n* = 48). Using a multimedia simulator or human models, the authors showed no differences in teaching image acquisition and interpretation skills to novice FAST examinees. It was concluded from this study that practical image acquisition skills acquired during simulated training could be directly transferred to human models. In addition, Bentley et al. [16] observed that the simulator might be a feasible alternative method for ultrasound education. 

Finally, Maloney et al. performed an interesting and highly applicable study on the role of simulation in teaching paramedics LUS in a pre-hospital setting—during varied ambulance driving conditions. They showed that the role of education and simulation with the use of POCUS has a significant impact on the quality of the images obtained and the time in their acquisition [17]. 

## 5. BLUE Protocol

To quickly identify and treat the cause of acute respiratory failure, in 2008, Lichtenstein introduced the BLUE (bedside lung ultrasound in emergency) protocol [18]. To date, it is the best known and most accurate of the existing LUS protocols. This study aimed to target diagnostic imaging to identify life-threatening conditions that can lead to acute respiratory failure. Lichtenstein’s task was to develop a protocol by which an ultrasound technician could perform a bedside ultrasound examination in a matter of minutes and implement the correct treatment based on six ultrasound probe positions. The examination is carried out in the chest’s upper (Figure 3), lateral, and posterolateral areas (Figure 4). Within the described six regions on both sides of the chest, the presence of characteristic ultrasound images and signs are assessed, among which are: the A (Figure 5) and B lines (Figure 6), lung sliding sign, lung point (Figure 7), tissue-like sign, and shred sign [18]. 

In an early 1998 study, Lichtenstein and Mezière examined 66 patients with dyspnea for the presence of B-line artefacts during the diagnosis of suspected pulmonary oedema. The authors found such a pattern in 100% of patients with pulmonary oedema, while it was absent in 92% of patients with chronic obstructive pulmonary disease (COPD) and 98.75% with a healthy lung [19]. A second study describing the validity of using bedside LUS during the differential diagnosis of dyspnea is that of Volpicelli et al. [20]. The authors evaluated the presence of B-lines in 300 patients hospitalized in the ED for dyspnea. They demonstrated a sensitivity of 87.5% and specificity of 97.7% in diagnosing alveolar-interstitial syndrome.

It is noteworthy that the previously cited study by Lichtenstein and Mezière [19] did not include a myocardial ultrasound assessment. Given that this is a significant causal group, there is justification for extending the BLUE protocol in this context. A straightforward tool for such assessment is the basic focus assessed transthoracic echocardiography (FATE) protocol. This protocol is based on traditional echocardiography but focused on identifying underlying abnormalities, including evaluation of the size of individual heart cavities, global systolic function, the presence of pericardial sac fluid, cardiac wall hypertrophy, or determination of the degree of inferior vena cava respiratory collapse [21,22].

The FATE protocol, using elements of echocardiography reduced to a simple ultrasound examination of a few minutes, aims at basic identification of the causes of hemodynamic abnormalities [23], which can have a tangible impact on the diagnosis of the causes of dyspnea. The entire examination according to the FATE protocol can often be found in the form of so-called “FATE Cards,” which are schematic protocols indicating how the projections are obtained and the pattern of significant pathologies.

Typical acoustic windows used for this type of study are the right parasternal, subcostal, apical, and left parasternal windows (Figure 8).

## 6. FALLS Protocol 

The need to diagnose the lungs and myocardium in terms of correlating existing concomitant pathologies was also noted by Daniel Lichtenstein in developing the FALLS (fluid administration limited by lung sonography) protocol. This protocol is a far less-known diagnostic tool than the BLUE protocol described earlier. The FALLS protocol is a dynamic imaging modality for the differential diagnosis of septic, hypovolemic, cardiogenic, and obstructive shock. 

The FALLS protocol diagnosing the causes of cardiogenic shock is based on Weil’s shock qualification criteria. First, we evaluate the presence of free fluid in the pericardial cavity, followed by features of right ventricular enlargement, ending with the evaluation of pleural sliding motion. The purpose of such diagnosis is to exclude cardiac tamponade, pulmonary embolism, and pneumothorax, which are conditions that can directly worsen heart failure, causing obstructive shock. Moreover, in the FALLS protocol, pulmonary oedema, which clinically corresponds to left-sided heart failure, is excluded or confirmed during the evaluation of B-line artefacts [24]. We also evaluate changes in inferior vena cava (IVC) diameter. These changes seem to be useful in terms of indications and contraindications for fluid therapy and the timing of its discontinuation [25,26].

Of particular note is the complementarity of the FALLS protocol with the BLUE protocol in diagnosing the causes of dyspnea and shock. [27]. High sensitivity (81%) and specificity (99%) of the BLUE protocol in the diagnosis of pulmonary embolism, as well as the usefulness of the diagnostic information derived from B-line observations in pulmonary oedema (sensitivity 97%, specificity 95%), should be emphasized [28]. 

## 7. FAST Protocol in the Context of the Respiratory System

The FAST protocol and its later modifications are commonly used to evaluate trauma patients. The name originated as an acronym for focused assessment with sonography for trauma. While the mortality and morbidity associated with trauma are not only due to damage to abdominal organs, a common complication of trauma is also pathologies within the thoracic organs, including the lungs. 

In the next decade, the term extended FAST (EFAST) was adopted, incorporating the differential diagnosis of pneumothorax and hemothorax. The examination in the EFAST protocol was extended to include the bilateral application of the ultrasound probe on the chest, among others, in the third to fifth intercostal space in the axillary–anterior line (Figure 9). The usefulness of using the EFAST protocol in the diagnosis of pneumothorax has been highlighted in many studies [29], repeatedly proving the superior sensitivity of ultrasound over the traditional physical examination of the chest [30]. 

Ultrasound diagnosis using the FAST protocol is not meant for the differential diagnosis of dyspnea itself but for the rapid diagnosis of life-threatening pathologies in the thoracic area, including pneumothorax and hemorrhage into the pleural cavities. Therefore, the authors of this article thought it appropriate to describe this protocol as a component of the ultrasound diagnostic elements of patients with dyspnea. 

## 8. Conclusions

Bedside ultrasonography, or point of care (POCUS), is developing rapidly; however, the knowledge about the use of LUS in a pre-hospital setting is scarce, highlighting the need for further research in this field. Additionally, despite the possibility of using various ultrasound protocols in diagnosing a patient with dyspnea, there is no comprehensive and, at the same time, highly sensitive and specific protocol covering a satisfactory saccade of differential diagnosis of this symptom. It seems reasonable to conduct further targeted research to create such a dedicated solution.

## Figures and Tables

**Figure 1 medicina-59-00224-f001:**
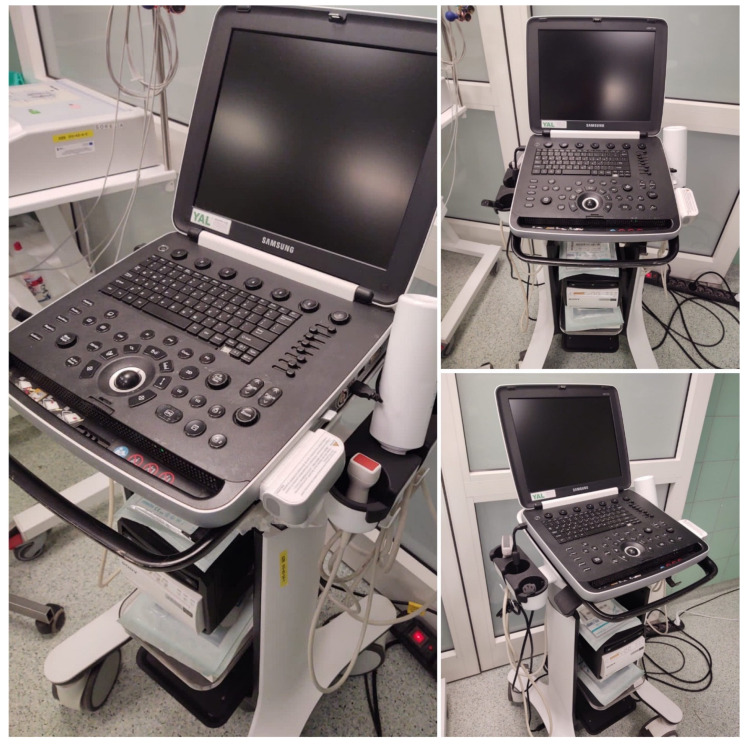
Stationary ultrasound machine (source: author’s material-DK).

**Figure 2 medicina-59-00224-f002:**
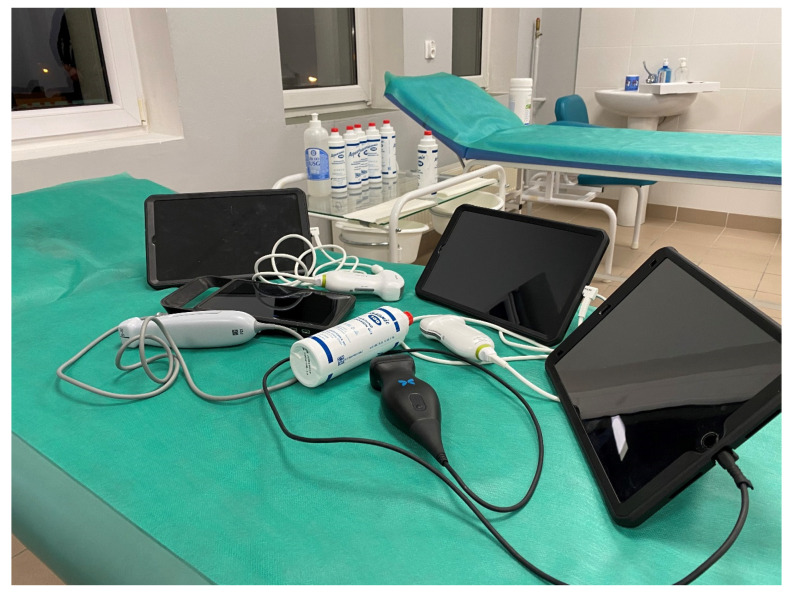
Mobile ultrasound equipment (source: author’s material-DK).

**Figure 3 medicina-59-00224-f003:**
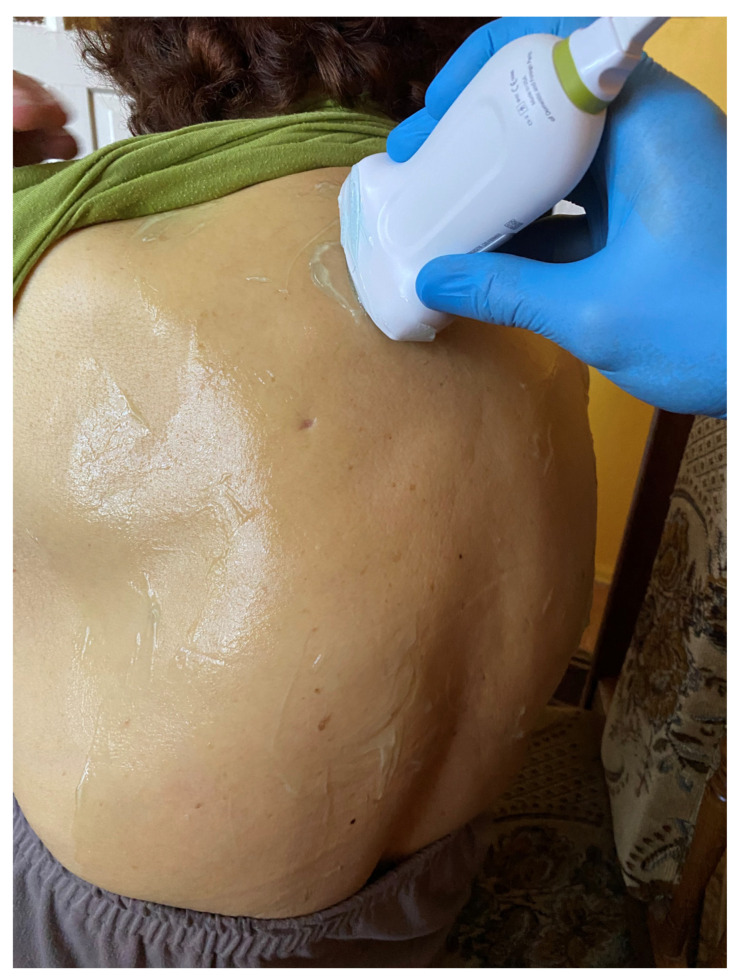
Lung examination according to the BLUE protocol—application of the head in the upper part of the chest (source: author’s material-DK).

**Figure 4 medicina-59-00224-f004:**
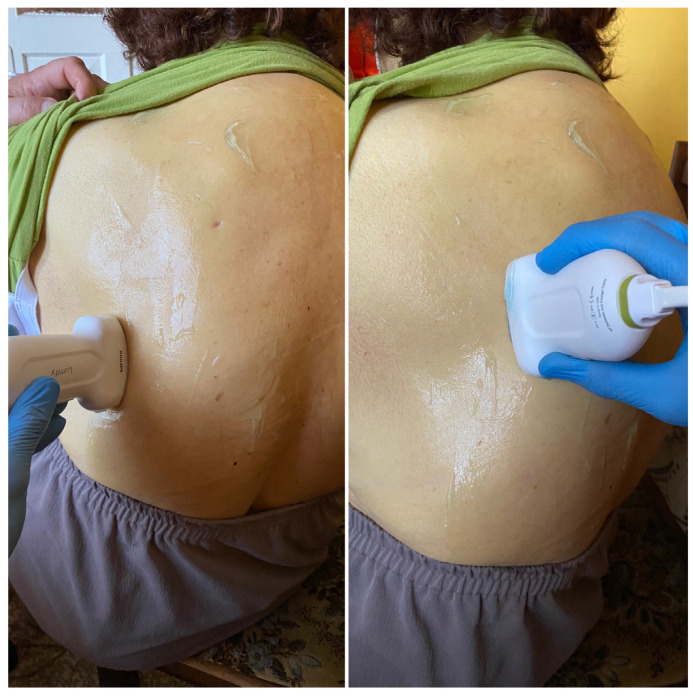
Lung examination according to the BLUE protocol—touchdown in the lateral and posterior–lateral parts of the chest (source: author’s material-DK).

**Figure 5 medicina-59-00224-f005:**
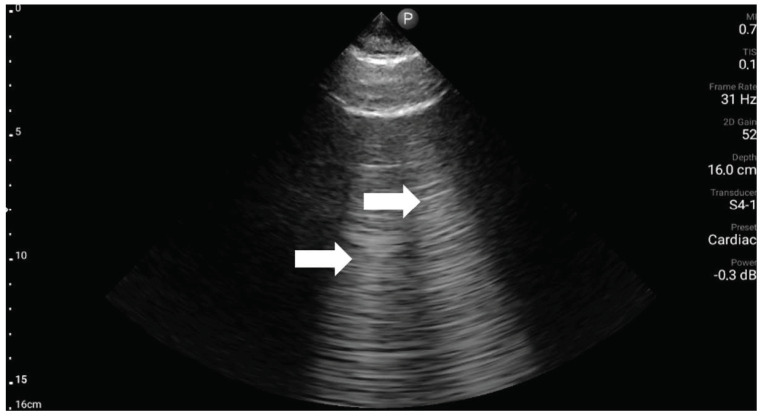
Ultrasound symptom “A-lines” (source: author’s material-DK).

**Figure 6 medicina-59-00224-f006:**
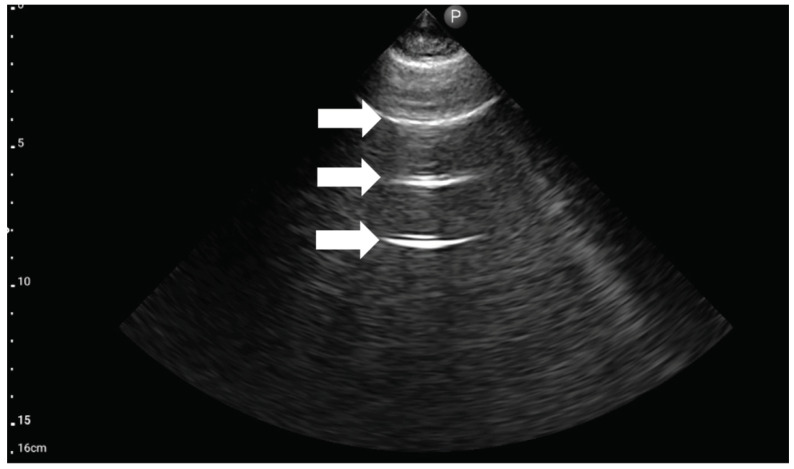
Ultrasound symptom “B-lines” (source: author’s material-DK).

**Figure 7 medicina-59-00224-f007:**
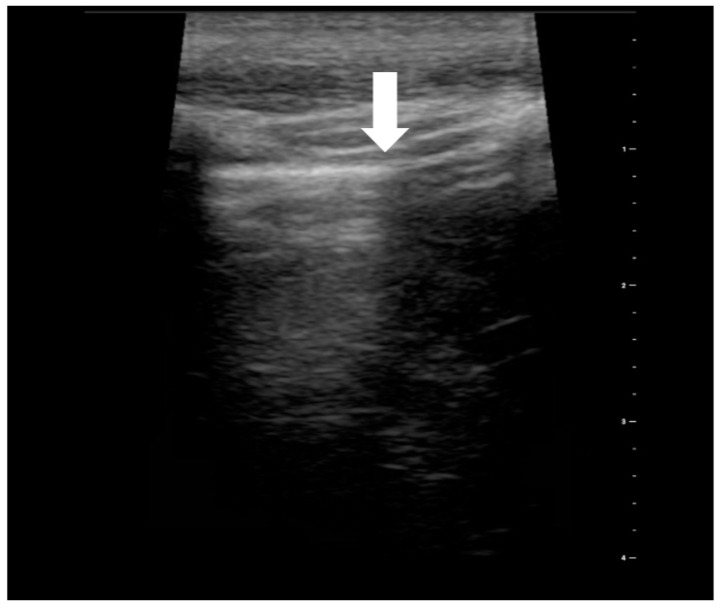
Ultrasound symptom “lung point” (source: author’s material-DK).

**Figure 8 medicina-59-00224-f008:**
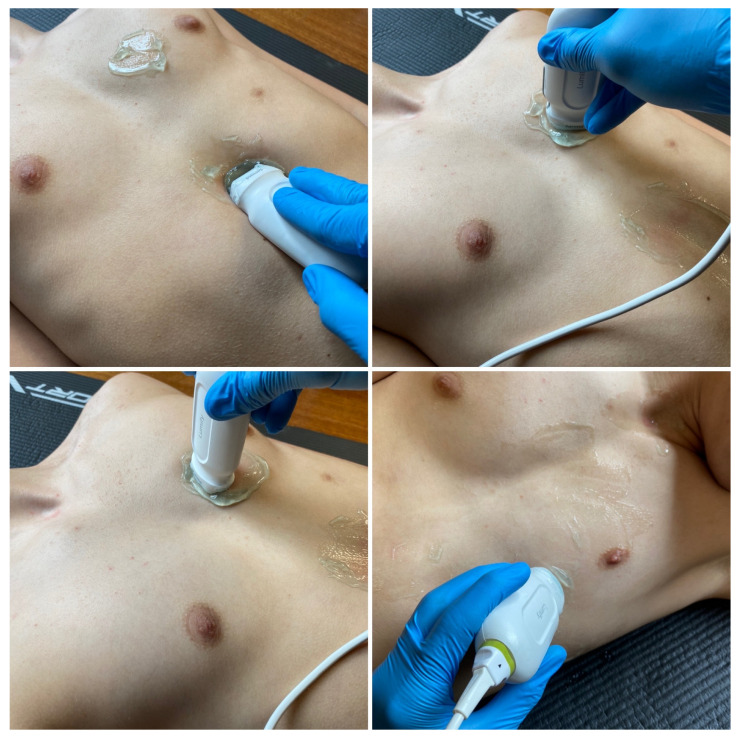
Acoustic windows are used during diagnosis according to the FATE protocol, i.e., subcostal, parasternal window in the short and long axis, and apical window (source: author’s material-DK).

**Figure 9 medicina-59-00224-f009:**
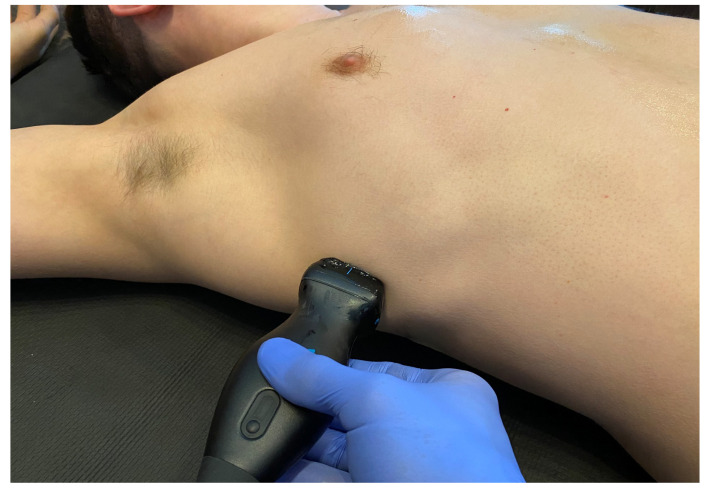
Application of the ultrasound camera in the anterior axillary line (source: author’s material-DK).

## Data Availability

Not applicable.

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
