# Peer review of "Concise, Practical Review on Transthoracic Lung Ultrasound in Prehospital Diagnosis of Dyspnea in Adults"

_medicina, 2023, doi:10.3390/medicina59020224_

Round 1

Reviewer 1 Report

       the manuscript is certainly interesting, and the topic is very relevant. There are potentially very valuable lessons to learn from this manuscript. However, many issues need to address before the manuscript is suitable for publication.
Overall:
1. There are many spelling errors in this manuscript. See guide for authors for a free grammar checker. E.g.……..etc. 

Title: - "Concise, Practical Review on Transthoracic Lung Ultrasound in 2 Prehospital Diagnosis of Dyspnea in Adults." Is a brief phrase describing the contents of the paper.
Abstract should be informative and completely self-explanatory, include the following structures: -
1. Introduction: Please describe the problem or lack of knowledge that address by this study

2. Aim of the study: please added the aims or purposes of the research and its relationship with other studies in the field.
3. Methods: Please add the study methodology that use in this study.

4. The result; - should be presented with clarity and precision
5. The conclusion should relate to the aim and problem that describe under introduction and aim
6. The keywords are following the abstract, use about five to 10 key words not mention in the title.

Introduction
1. The introduction is ideally in structures of the following order: -

Background information on the topic including incidences of the problem, definitions, a description of a problem or a lack of knowledge on a certain topic, a segment on WHY this is a problem and WHY this study is necessary, and a segment that underlines the research question that should be answered (based on the problem describe earlier) please rearrange again.

2. The aim of the research: - What lessons were learned? What policies were changed?
3. What is the hypothesis of the study? Is the hypothesis clear?

Methods
1.The methods section should write in a way that everyone could repeat this study in the same manner.
3. You should add study question, primary objectives and secondary objectives if available.

Results
1. Results are meaningless without a better description in figures. So please add this. This is the core of the manuscript and the most valuable.
Discussion
1.A discussion should add offer a short overview of different protocols, and an in depth discussion of the interpretation of them.
3. The last paragraphs of the discussion are very relevant. Please add some suggestions on how such problems can improve in your own hospital. How did you manage other problems? Share some solutions that applied after this research.

Conclusion:
is logic in manner and relevant.

References:
1. Please use uniform references, when available with DOI.

2. Make sure update the old references

Author Response

Dear Reviewer,

Thank you for your valuable input. We have now improved the quality of the manuscript with language edition by a native speaker. We have implemented some improvements accrding to the quidelines provided which are now listed below. Howerver, we would like to point out that this is not an original research paper but a narrative review of literature, therefore no study methodology can be provided and no results are presented. In a narrative review there is no predefined search strategy, as this is not a systematic review, nor is there a research question (https://www.sciencedirect.com/topics/psychology/narrative-review#:~:text=A%20narrative%20review%20is%20the,only%20a%20topic%20of%20interestA literature review was conducted with  the Pubmed and Scopus databases using the key words "POCUS", “lung ultrasound”, “emergency ultrasound”, “paramedic ultrasound” and “prehospital ultrasound”. Literature regarding point of care ultrasound of lungs was reviewed by the authors and presented in a narrative review style. 

We have included many figures depicting the position of US probe and US image obtained, however no graphs or charts are presented in this manuscript. 

We added information on the methodology and changed the bibliography.

Thank you very much for your review, your time and valuable tips on editing the manuscript.

Yours faithfully

Damian Kowalczyk

Reviewer 2 Report

In contrast to abdominal and cardial sonography, lung ultrasound is POCUS. You have it excellent worked out. Scientific soundness could be improved.

Author Response

Dear Reviewer

Thank you for your valuable input. We have now improved the quality of the manuscript with language edition by a native speaker. We added information on the methodology and changed the bibliography. Thank you very much for your review, your time and valuable tips on editing the manuscript

Yours faithfully

Damian Kowalczyk

Reviewer 3 Report

Dear Authors

Thanks for this review of lung ultrasound in prehospital diagnosis of dyspnea patients. Authors discuss the different protocols. I think this manuscript is well written and interesting for readers. 

Author Response

Dear Reviewer

Thank you very much for your valuable comments and for taking the time to review our manuscript.

Yours faithfully

Damian Kowalczyk

Round 2

Reviewer 1 Report

the manuscript is certainly interesting, and the topic is very relevant. There are potentially very valuable lessons to learn from this manuscript. However, the manuscript is suitable for publication.

Title: - Concise, Practical Review on Transthoracic Lung Ultrasound in 2 Prehospital Diagnosis of Dyspnea in Adults.